# Effect of tetanic stimulation prior to train-of-four monitoring on the time to muscle response stabilization with neuromuscular blockade with rocuronium in patients aged 60 to 80 years: A prospective randomized controlled trial

**Angela Henrique Silva Ribeiro**[1]*, **Nubia Verçosa**[2], **Estêvão Luiz Carvalho Braga**[3], **Angelo Jorge Queiroz Rangel Micuci**[4], **Paulo Alípio Germano-Filho**[3], **Ismar Lima Cavalcanti**[1]

1 Department of General Specialized Surgery, Postgraduate Program in Medical Science, Universidade Federal Fluminense (UFF), Niterói, RJ, Brazil, 2 Department of Surgery, Postgraduate Program in Surgical Sciences, Universidade Federal do Rio de Janeiro (UFRJ), Rio de Janeiro, RJ, Brazil, 3 Department of General and Specialized Surgery, UFF, Niterói, RJ, Brazil, 4 Department of Anesthesiology, Hospital Federal de Bonsucesso, Rio de Janeiro, RJ, Brazil

☯ These authors contributed equally to this work.
* angelahenriqueanest@gmail.com

## Abstract

### Purpose

To determine the effect of tetanic stimulation on the time to achieve stabilization of the T1 height, by acceleromyography train-of-four (TOF) ratio monitoring, in patients aged 60 to 80 years.

### Methods

Randomized, prospective, single-blind, controlled trial in patients aged 60 to 80 years under-going elective surgery under general anesthesia in two university hospitals in Brazil between December 2019 and March 2022. This trial was performed in accordance with the principles of the Helsinki Declaration and the guidelines for good clinical research practice in pharma-codynamic studies of neuromuscular blocking agents II. Participants were randomly allo-cated by computer-generated numbers to receive tetanic stimulation (intervention group, n = 25) or not receive tetanic stimulation (control group, n = 25) to achieve T1 height stabili-zation on TOF ratio monitoring. The primary endpoint was to evaluate the effect of tetanic stimulation on the time to achieve stabilization of the T1 height. We also evaluated the effect on the onset time of rocuronium and time to recover from neuromuscular blockade with a single dose of rocuronium 0.6 mg.kg$^{-1}$ with spontaneous recovery.

**Data Availability Statement:** The minimal dataset is held in a public data repository at Mendeley Data. Data identification number: DOI: 10.17632/fyhpfpk5cm.1 Direct URL to data: https://data.mendeley.com/datasets/fyhpfpk5cm

**Funding:** The author(s) received no specific funding for this work.

**Competing interests:** The authors have declared that no competing interests exist.

## Results

There was no statistically significant difference between the groups in the time to T1 height stabilization (intervention: 4.3 [2.0–8.0] min; control: 5.8 [2.0–10.5] min; p = 0.202), onset time of the neuromuscular blocking agent (intervention: 1.9 [±0.7] min; control: 2.2 [±1.0] min; p = 0.219), or neuromuscular blockade recovery times to normalized TOF values of 0.7 (intervention: 106.1 [±37.2] min; control: 91.7 [±27.5] min; p = 0.295), 0.8 (intervention: 114.3 [±40.1] min; control: 97.8 [±28.9] min; p = 0.225), 0.9 (intervention: 126.5 [±44.0] min; control: 106.9 [±30.6] min; p = 0.153), and 1.0 (intervention: 138.3 [±43.4] min; control: 123.8 [±43.5] min; p = 0.425).

## Conclusion

The results of this study suggest that the recommendation to use tetanic stimulation to shorten the time to T1 height stabilization in neuromuscular monitoring did not alter the stabilization time of the T1 twitch or the pharmacodynamics of neuromuscular blockade with rocuronium in patients aged 60 to 80 years.

## Introduction

### Background

Neuromuscular blocking agents (NMBAs) are widely used in healthcare facilities worldwide. During general anesthesia, they improve surgical conditions and facilitate tracheal intubation, and thereby maintain controlled mechanical ventilation in intensive care [1]. Neuromuscular blockade (NMB) monitoring reduces complications [2] and hospital costs. Therefore, it should be routinely employed [3], and it is part of the safe surgery checklist [4]. Objective monitoring of NMB is the only reliable way to ensure the effectiveness of NMB and the safety of its reversal [5]. The recommended method for quantitative monitoring of stimulation patterns is the train-of-four (TOF) by acceleromyography, which assesses the level of muscle strength according to the ratio of the fourth (T4) to first muscle twitch (T1) in the TOF sequence (TOF ratio = T4/T1). The endorsed assessment is the monitoring of the adductor pollicis muscle in response to ulnar nerve stimulation [6].

The Stockholm Revision Guidelines are guidelines for good clinical research practice in pharmacodynamic studies of NMBAs II, intended to standardize researches on NMB so that the results can be extrapolated to clinical practice. According to the guidelines, before NMBA injection, the device should be calibrated to adjust the gain or current intensity to obtain a T1 height of 100%. For research purposes, the muscle response to the electrical stimulus should be stabilized for reliable and accurate data interpretation [7]. According to the guidelines, stabilization is defined as a maximum variation of 5% in the T1 height (first contraction measured by acceleromyography) for at least 2 minutes before injection of NMBAs [7]. However, it may take up to 20 minutes to achieve a stable response, which is too long in clinical practice. To decrease the stabilization period, 50 Hz tetanic stimulation was recommended before calibration in adult patients [8].

The time to achieve stabilization of the control response may influence the duration and onset time of the NMB [9]. The pharmacokinetics and pharmacodynamics of NMBAs can be altered by aging [10]. Patients aged over 60 years old were not included in the Stockholm

guidelines [7]. We hypothesized that the tetanic stimulation could decrease stabilization time in this population.

### Objectives

The primary outcome of this randomized controlled trial was to determine the time to stabilize the muscle response using tetanic stimulation prior to TOF stimulation in patients aged 60 to 80 years. Secondary outcomes were the initial and final T1 height, the onset of action and clinical duration of rocuronium, and the NMB recovery time with and without tetanic stimulation in this age group.

## Materials and methods

### Study design

We conducted a multicenter, prospective, randomized, single-blind, controlled trial with equal randomization (1:1 allocation ratio for two groups). The study was conducted in Rio de Janeiro, Brazil (2 sites). No changes to the anesthesia protocol or methods were made after trial commencement.

### Population

The inclusion criteria were either sex, aged 60 to 80 years old, an American Society of Anesthesiologists (ASA) physical status classification I to III, a body mass index (BMI) between 18.5 and 29.9 kg/m$^2$ and elected for surgery with an expected operative time exceeding 60 minutes.

The exclusion criteria were: presence of neuromuscular disorders; renal or hepatic dysfunction; use of furosemide, aminoglycosides, corticosteroids, lithium, statins, anti-epileptics, aminophylline, azathioprine, cyclophosphamide, anti-inflammatory drugs, and magnesium; allergy to the drugs used in the study; and history or predictors of a difficult airway.

### Setting

This trial was conducted between December 2019 and March 2022 in two tertiary university general hospitals located in Rio de Janeiro, Brazil. Recruitment started on December 19th, 2019, at Gaffrée & Guinle Federal University Hospital and on July 29th, 2021, at Pedro Ernesto University Hospital. Recruitment ended on March 17th, 2022, in both hospitals.

### Ethics

The original study protocol was approved on August 8th, 2019, by the Institutional Ethics Committee of Gaffrée & Guinle Federal University Hospital (original approval document: CAAE number 03260918.4.0000.5258; opinion number 3.494.496). An amendment was made to the 2019-approved study protocol after commencement of the trial to request the inclusion of a co-participating institution (Pedro Ernesto University Hospital) to increase the study sample due to the COVID-19 pandemic. The amendment was approved on May 25th, 2021, by the Institutional Ethics Committee of Gaffrée & Guinle Federal University Hospital (amendment approval document: CAAE number 03260918.4.0000.5258; opinion number 4.733.353). The Institutional Ethics Committee of Pedro Ernesto University Hospital approved the study protocol on July 29th, 2021 (approval document: CAAE number 03260918.4.3001.5259; opinion number 4.872.130). This study was registered on the Brazilian Clinical Trials Registry (ReBEC) under number RBR-35msdt, and written informed consent was obtained from all 60 patients prior to surgery. Data were stored in the Mendeley Database Repository. Dataset data presented in this paper are provided as S1 Dataset. This manuscript adheres to the applicable

CONSORT guidelines S1 Checklist and follows the Stockholm Revision guidelines [7]. The clinical study protocol is provided as S1 Protocol.

## Randomization and blinding

The randomization sequence was created using WinPepi (Program for Epidemiologists for Windows), where 60 random numbers were generated and stored in sealed and opaque envelopes by an investigator with no clinical involvement in the trial. The envelopes were opened by a trial researcher after the induction of general anesthesia. The patients were randomly allocated to one of two groups in a 1:1 ratio: the intervention group received tetanic stimulation (n = 30) to achieve signal T1 height stabilization during TOF monitoring, while the control group did not receive tetanic stimulation (n = 30), obeying the random allocation sequence. The participants were blinded to the allocation, as they were under anesthesia. A single researcher who was aware of the group allocations collected the data. The dataset was unblinded for analysis once data collection ended.

## Anesthesia and monitoring

Patients were evaluated, and the following complementary tests and data were performed and collected preoperatively: complete blood count, clotting tests, urea, creatinine clearance, sodium, potassium, glucose, liver function tests, chest radiography, and electrocardiography.

In the operating room, peripheral venous access with an intravenous catheter was established on the arm contralateral to the side of neuromuscular monitoring. Patients were monitored using a multiparameter monitor that included cardioscopy, pulse oximetry, capnometry, capnography, esophageal and peripheral temperature, noninvasive blood pressure in the lower extremities (GE Dash 4000 multiparametric monitor, GE healthcare), and bispectral index (BIS) monitoring (BIS® sensor, Aspect Medical Systems Inc., Newton, MA, USA). The lower extremities of all the patients were covered with a warming blanket (Bair Hugger, 3 M United, MN, USA), and the core body temperature was maintained above 36°C.

The skin was cleaned, and two pediatric electrodes were placed over the ulnar nerve path, near the wrist, to measure the acceleration of the adductor pollicis muscle in response to ulnar nerve stimulation. Preload was applied using a hand adapter (Hand Adapter®, Organon Oss, The Netherlands) to decrease signal variability, and a piezoelectric crystal transducer (TOF-Watch SX® acceleromyograph, Organon Ltd., Dublin, Ireland) was firmly attached to the ventral distal phalanx of the thumb. The temperature sensor was placed over the hypothenar eminence of the monitored hand. The arm and fingers (except the thumb) were immobilized in the extended position with surgical tape throughout the procedure. All the data obtained via the device's software (TOF-Watch SX® Monitor Version 2.5.int, 2007) were stored in a personal computer.

After preoxygenation (6 L.min$^{-1}$) for 3 minutes using a face mask, total intravenous anesthesia was induced and maintained until the end of surgery with continuous intravenous target-controlled infusion of propofol (2 to 4 µg.ml$^{-1}$), guided by the BIS (40 to 60), using an Agilia®TIVA infusion pump (Fresenius-Kabi, Louviers, France) with the Marsh model. Remifentanil (0.1 to 0.3 µg.kg$^{-1}$.min$^{-1}$) was also continuously infused with the Agilia®TIVA infusion pump.

At an appropriate anesthesia depth (BIS index below 60), TOF stimulation of the ulnar nerve was started for 1 minute. In the control group, calibration was performed to achieve supramaximal stimulation (the electrical stimulus was automatically applied by the device at 15% to 20% above that necessary for a maximal response) [7] using the built-in calibration function (CAL 2) of the device to obtain a twitch height of 100% of T1, yielding the

supramaximal current (thus ensuring that all muscle fibers were activated). We then waited for signal stabilization, which is defined as a maximum variation of 5% of the T1 height for at least 2 minutes. When the T1 height variation was greater than 5%, subsequent calibrations were performed until signal stabilization was achieved. In the intervention group, after starting TOF stimulation, 50 Hz tetanic stimulation was applied for 5 seconds, and then the TOF sequence was started and calibrated as in the control group. The TOF pattern was a supramaximal square wave stimulus of 200 ms duration, 2 Hz frequency, and a 15-second interval between consecutive TOF stimuli (Fig 1).

Neuromuscular monitoring was performed with a TOF-Watch SX® acceleromyograph (Organon Ireland Ltd., Dublin, Ireland). TOF stimulation was used, and the TOF ratio was calculated by comparing the magnitude of the fourth evoked response (twitch) of the adductor pollicis muscle with that of the first response (T4/T1). The height of muscle contraction

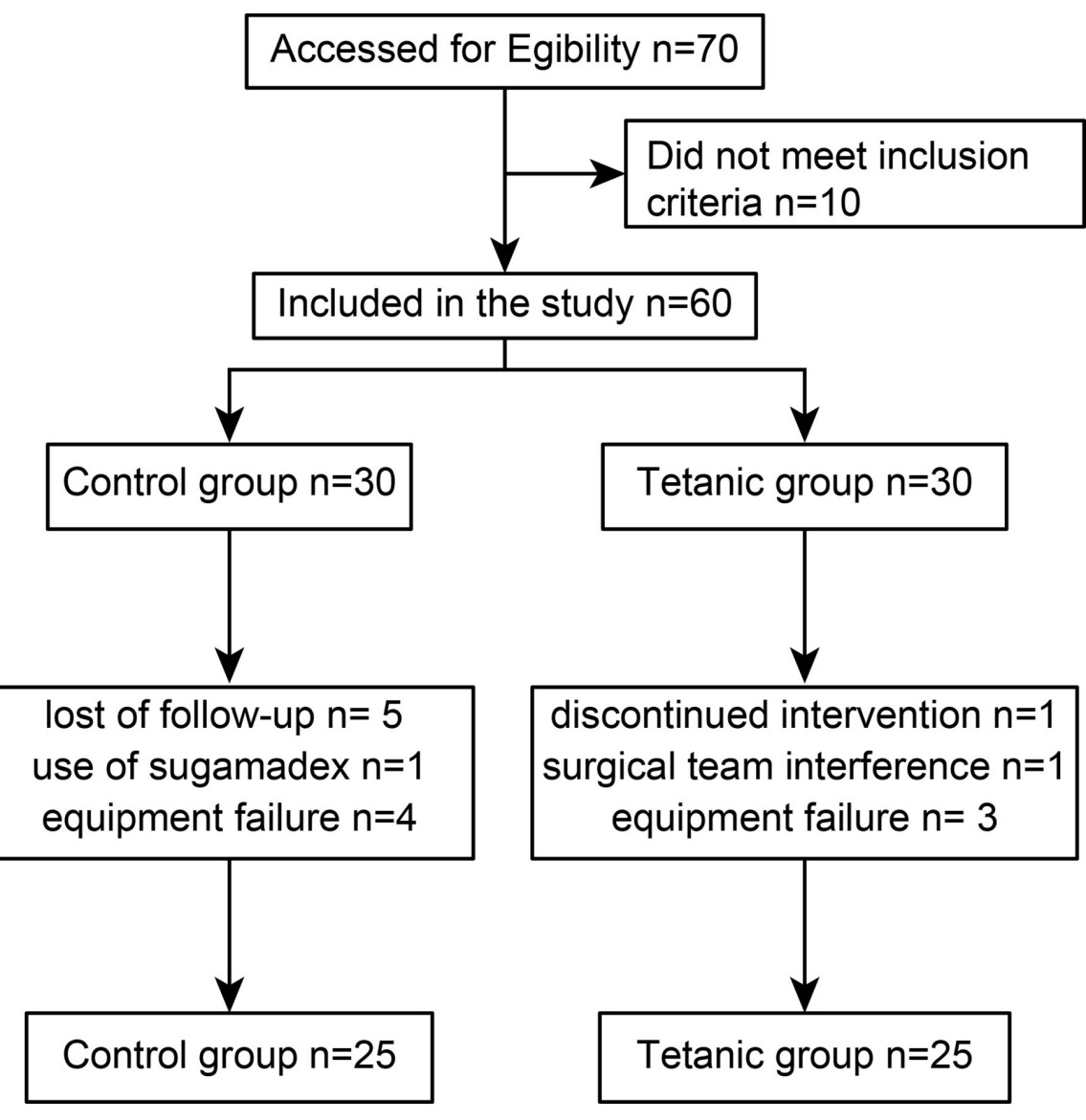

**Fig 1. CONSORT flow diagram.**

at the first twitch of TOF (T1) was also measured [11]. An interfaced personal computer registered and recorded the effect of four stimuli (TOF sequence: T1, T2, T3, and T4) from the TOF-Watch SX® monitor program (version 2.5. Int, 2007, Organon Ltd., Dublin, Ireland).

Rocuronium 0.6 mg.kg⁻¹ was administered intravenously as a single bolus within 5 seconds after the stabilization period. When the TOF count was 0 for three sequential measurements, the trachea was intubated with a cuffed tube, and mechanical ventilation initiated.

Before the end of the surgical procedure, nalbuphine 0.1 mg.kg⁻¹ and 30 mg ketorolac were administered intravenously for postoperative analgesia. Intravenous dexamethasone 4 mg and ondansetron 8 mg.kg⁻¹ were administered as antiemetic therapy. NMB reversal was spontaneous according to acceleromyography monitoring. At the end of surgery, the patients were extubated when the TOF value was 1.0 (100%, normalized) and then transferred to the postanesthesia care unit.

The baseline acceleromyographic TOF values after calibration are often above 100%, so a TOF ratio of 0.9 at the time of NMB reversal does not always translate into recovery from NMB. Normalization of the TOF value is therefore recommended, consisting of relating the TOF values obtained at NMB recovery with the baseline value (the first TOF value obtained after calibration) [12]. No complications were reported postoperatively regarding NMB. All recovery values described in this study were normalized, as recommended by the guidelines for good clinical research practice in pharmacodynamic studies of NMBAs II [7].

## Outcomes

The primary endpoint was the time to achieve T1 height stabilization in minutes in both groups (with and without tetanic stimulation) using the acceleromyography TOF-Watch SX® monitor. Secondary endpoints included the time to NMBA onset of action (time measured from rocuronium injection to 95% T1 height depression) and NMB recovery times with or without the interference of tetanic stimulation. No changes to the trial outcomes were made after trial commencement.

## Power calculation and sample size

Due to the paucity of data in the literature, the sample size was calculated based on a pilot study with 10 patients. Considering a significance level of 5% and a power of 80% to detect a reduction in the time to stabilization of the expected T1 height within 8 minutes after the tetanic stimulation, an estimated standard deviation (SD) of 10 minutes (minimum effect size of 0.8 SDs) and an estimated 20% of potential losses, a sample size of 30 patients per group was needed.

## Statistical analysis

The results are reported as absolute and relative frequencies (percentages) for categorical variables or measures of central tendency and dispersion (mean [SD] or median [IQR]) for numerical variables. Student's *t* test was used to compare normally distributed continuous variables (age, BMI, and creatinine clearance) between the intervention and control groups. For categorical variables, Pearson's chi-square test or Fisher's exact test was used to compare proportions (sex and ASA class) between the intervention and control groups. Due to the different outcomes studied, multivariate analysis of variance (MANOVA) with Bonferroni adjustment was applied in the comparison between groups to control for type I error (time to stabilization, onset time, initial and final T1 height, clinical duration, and NMB recovery variables). For variables with asymmetrical distributions, logarithmic transformation was used. A p value less

than 0.05 was considered statistically significant (p < 0.05). Statistical analysis was performed using SPSS version 28.0 (IBM Corp, Chicago, IL, USA).

## Results

During the study period, 60 eligible patients were recruited and randomized, and 50 patients completed the study (Fig 1). Because the result of the intervention is immediate, not prolonged, we did not consider the intention-to-treat population in the analysis for the primary outcome. There were no missing values.

Participant characteristics did not differ between the two groups (Table 1). In the control group, creatinine clearance was not obtained in two participants, the monitor interface failed during the procedure in two participants, and NMB reversal with sugammadex was required in one participant due to a short operation time. In the intervention group, the surgical team interfered by manipulating the monitored arm in one participant, monitoring was discontinued in one participant due to surgery cancellation, and the interface failed during monitoring in three participants.

There was no statistically significant difference in the time to stabilization of the first muscle response to tetanic stimulation (intervention: 4.3 [2.0–8.0] min; control: 5.8 [2.0–10.5] min; p = 0.202) or the NMBA onset time (intervention: 1.9 [±0.7] min; control: 2.2 [±1.0] min; p = 0.219) between the groups (Table 2).

There were no significant between- or within-group differences in the initial and final T1 height values (Table 3).

The groups did not significantly differ in the recovery times: time to reappearance of two TOF responses (p = 0.213), clinical duration (p = 0.300), time to reappearance of T1 75% (p = 0.629), or NMB recovery times to normalized TOF values of 0.7 (intervention: 106.1 [±37.2] min; control: 91.7 [±27.5] min; p = 0.295), 0.8 (intervention: 114.3 [±40.1] min; control: 97.8 [±28.9] min; p = 0.225), 0.9 (intervention: 126.5 [±44.0] min; control: 106.9 [±30.6] min; p = 0.153), or 1.0 (intervention: 138.3 [±43.4] min; control: 123.8 [±43.5] min; p = 0.425) (Table 4).

**Table 1. Participant characteristics.**

| Variable | Intervention group (n = 25) | Control group (n = 25) |
|---|---|---|
| Age (years), mean ± SD (95% CI) | 66.7 ± 5.4 (64.7–68.7) | 68.1 ± 5.9 (65.7–70.6) |
| Sex, n (%) | | |
| Male | 13 (52.0) | 17 (68.0) |
| Female | 12 (48.0) | 8 (32.0) |
| BMI (kg/m$^2$), mean ± SD (95% CI) | 26.0 ± 3.2 (24.6–27.3) | 24.6 ± 3.2 (23.2–25.9) |
| ASA class, n (%) | | |
| I | 4 (16.0) | 3 (12.0) |
| II | 15 (60.0) | 17 (68.0) |
| III | 6 (24.0) | 5 (20.0) |
| Creatinine clearance (mL/min) | 75.9 ± 15.7 | 75.0 ± 18.7 |

ASA: American Society of Anesthesiologists; BMI: body mass index; CI: confidence interval; SD: standard deviation.

[*] Student's *t* test.

[**] Pearson's chi-square test.

**Table 2. Time to stabilization and onset time.**

| Variable | Intervention group (n = 25) | Control group (n = 25) | p value[*] |
|---|---|---|---|
| Time to stabilization[a] (min) | 4.3 (2.0–8.0) | 5.8 (2.0–10.5) | 0.202 |
| Mean ± SD | 4.8 ± 0.9 | 6.9 ± 0.8 | |
| 95% CI | 3.1–6.6 | 5.2–8.5 | |
| Onset time[b] (min) | 1.9 ± 0.7 | 2.2 ± 1.0 | 0.219 |
| 95% CI | 1.6–2.2 | 1.8–2.6 | |

CI: confidence interval.

[*] Multivariate analysis of variance (MANOVA).

[a] Described as the median (25th-75th percentiles).

[b] To 95% T1 depression. Described as the mean ± SD. For the time to stabilization, logarithmic transformation was used.

## Discussion

This randomized, single-blind, controlled trial analyzed the time to achieve T1 height stabilization with TOF stimulation, and the onset of action and recovery times after a single dose of rocuronium in patients aged 60 to 80 years. The results demonstrated that the use of tetanic stimulation produced no statistically significant difference in the time to achieve a stable response at T1 height in this population, although it produced less variation in the stabilization time (Table 2). The mean time to stabilization was lower, 4.3 minutes, in the intervention group, which did not differ significantly from the mean 5.8 minutes of the control group (Table 2).

According to the guidelines for good clinical research practice in pharmacodynamic studies of NMBAs II [7], 50 Hz tetanic stimulation using the TOF-Watch SX® device is recommended to shorten the time to T1 height stabilization to 2–5 minutes when using TOF stimulation for research purposes, based on studies with adults and young participants.

The stimulation pattern and time to achieve a stable response can alter the NMB onset and recovery times. Kopman et al. [13], in a study on younger adults, found that tetanic prestimulation reduced T1 twitch variability and stabilization time. They explained that this was due to the posttetanic potentiation period in the tetanic stimulation group, liberating a large amount of neurotransmitters (acetylcholine) in the neuromuscular junction. We did not find a reduction in the stabilization period but did identify large T1 height variability in both groups, but the variability was lower in the tetanic prestimulation group (Table 2).

McCoy et al. [9] investigated the use of atracurium, mivacurium, and vecuronium in adult patients. They associated the duration of nerve stimulation before injection of the neuromuscular agents with the variability in responses, including the onset of action and clinical duration of NMB. They found faster onset of action and longer NMB recovery times with increasing periods of stimulation, speculating that prolonged stimulation increased blood

**Table 3. Initial and final T1 height in each group.**

| Variable | Intervention group, m ± SD (95% CI) (n = 25) | Control group, m ± SD (95% CI) (n = 25) | p value[*] |
|---|---|---|---|
| Initial T1 height (%) | 93.8 ± 11.8 (88.9–98.6) | 94.1 ± 6.8 (91.3–96.9) | 0.978 |
| Final T1 height (%) | 98.9 ± 24.5 (88.8–109) | 98.8 ± 20.4 (90.3–107.2) | 0.886 |
| Difference | 5.2 (−5.6 to 15.9) | 4.7 (−4.3 to 13.6) | 0.884 |

CI: confidence interval; SD: standard deviation; T1: first train-of-four twitch.

[*] Between-group comparison using multivariate analysis of variance (MANOVA). [**] Within-group comparison using Student's *t* test for paired samples.

**Table 4. Clinical duration and neuromuscular blockade recovery variables.**

| Variable | Intervention group, m ± SD (95% CI) (n = 25) | Control group, m ± SD (95% CI) (n = 25) | p value[*] |
|---|---|---|---|
| Rocuronium duration 95% T1 | 104.9 ± 39.4 (88.7–121.2) | 97.9 ± 32.0 (84.7–111.1) | 0.455 |
| Two TOF responses | 60.2 ± 24.0 (50.3–70.1) | 51.4 ± 16.3 (44.7–58.1) | 0.213 |
| Clinical duration of 25% T1 height | 65.9 ± 23.9 (56.1–75.8) | 57.5 ± 20.4 (49.1–65.9) | 0.300 |
| Time to achieve 75% T1 height | 90.7 ± 36.3 (75.7–105.7) | 82.6 ± 31.1 (69.8–95.5) | 0.629 |
| Time to a normalized TOF value of 40% | 87.1 ± 33.5 (73.3–100.9) | 75.9 ± 24.3 (65.9–85.9) | 0.329 |
| Time to a normalized TOF value of 50% | 92.4 ± 34.7 (78.1–106.8) | 81.1 ± 26.2 (70.3–91.9) | 0.368 |
| Time to a normalized TOF value of 60% | 98.8 ± 35.6 (84.1–113.5) | 85.6 ± 26.6 (74.6–96.5) | 0.302 |
| Time to a normalized TOF value of 70% | 106.1 ± 37.2 (90.7–121.4) | 91.7 ± 27.5 (80.3–103.0) | 0.295 |
| Time to a normalized TOF value of 80% | 114.3 ± 40.1 (97.8–130.9) | 97.8 ± 28.9 (85.9–109.7) | 0.225 |
| Time to a normalized TOF value of 90% | 126.5 ± 44.0 (108.4–144.7) | 106.9 ± 30.6 (94.3–119.5) | 0.153 |
| Time to a normalized TOF value of 100% | 138.3 ± 43.4 (118.6–158.0) | 123.8 ± 43.5 (105.9–141.7) | 0.425 |

CI: confidence interval; SD: standard deviation; T1: first TOF twitch; TOF: train-of-four.

[*] Multivariate analysis of variance (MANOVA).

flow, which would increase the delivery of the neuromuscular agents to the neuromuscular junctions. In our study, those differences were not significant between the groups, with a mean onset time of 1.9 (±0.7) minutes in the intervention group and 2.2 (±1.0) minutes in the control group (p = 0.219) and similar recovery times in both groups. As expected, we also found considerable variability in the TOF response and T1 height in both groups. Bjerring et al. [14] showed a longer onset time and duration of action of rocuronium in elderly patients when compared with younger patients, hence reemphasizing the importance of this fact increasing the risk of postoperative complications in this age group.

Although our sample was more heterogeneous than that in the investigation by Carlos et al. [15], our results are consistent with those that found no significant difference in the time to achieve T1 height stabilization [C: 195.0 (±203.0), T: 116.0 (±81.6)] or in neuromuscular agent onset time regardless of the use of tetanic stimulation in children. Our findings may indicate that the use of tetanic stimulation to shorten the time to T1 height stabilization is also not effective in the population aged 60 to 80 years. In their study, the use of tetanic stimulation in children decreased the recovery times to normalized TOF values of 0.7, 0.8, and 0.9. They explained the early recovery times in children because the calibration in the tetanic stimulation group coincided with the posttetanic period. In the current study, there was no change in recovery times when comparing the groups with and without tetanic stimulation. This difference can be explained because posttetanic potentiation occurs in young and healthy individuals, which maintains calcium reserve capacity in the sarcoplasmic reticulum of muscle fibers released after a tetanic stimulus and temporarily increases muscle strength [16]. However, in elderly individuals, this capacity may be compromised due to a reduction in muscle mass and neuromuscular responsiveness, with loss in motoneurons and morphological damage in neuromuscular junctions, altering acetylcholine receptors, which may affect posttetanic potentiation [17]. Aging also affects the functional and electrophysiological properties of the peripheral neural system, decreasing nerve conduction velocity, muscle strength and endoneural blood flow [18].

We confirmed that the TOF ratio is not affected by tetanic stimulation in this population, as demonstrated by Unterbuchner et al. [19] with children, ensuring safety in this quantitative monitoring for evaluating NMB recovery.

For safe extubation without postoperative complications, the literature recommends a minimum TOF value of more than 90% [20] to ensure that there are no complications from

residual NMB [21]. Baseline values and normalization ensure accurate acceleromyography, preventing misinterpretation of NMB recovery by ensuring a real TOF ratio above 0.9.

TOF-Watch SX® monitoring is vulnerable to movements and interference, so calibration and stabilization of the signal is essential for accuracy. The device is portable and easy to use, and it comes with software that registers in numbers and graphically the neuromuscular monitoring. This helps in the research field, as shown in a previous study using a personal computer to obtain those registries [22].

NMBAs act by competitively binding to nicotinic acetylcholine receptors at the neuromuscular junction [23]. Individuals over 60 years of age often have sarcopenia and decreased muscle strength. In addition to a decrease in muscle mass and an increase in body fat, aging is associated with reduced muscle fiber numbers and sizes [20]. These quantitative and qualitative changes in skeletal muscle can affect the interpretation of data in neuromuscular monitoring [24] but are still reliable. The proportion of aging adults in the population is increasing, and the elderly population is susceptible to the occurrence and consequences of postoperative residual blockade. It is necessary to better understand what happens when using NMB in this age group.

## Limitations

This study has limitations that should be noted. First, patients in this age group with a high BMI were not included, as we followed the Stockholm guidelines. Second, the preload that increases the accuracy and precision of acceleromyography [12] was never tested in older patients. Third, we had a dropout rate of 20%, the sample size was only calculated for the primary outcome, and the analysis population, as per protocol, resulted in anticonservative test decision. If we had achieved an effect size of 0.8 SDs, we would have found statistical significance, but the effect size found in the study was 0.5 SDs, making the study underpowered.

## Conclusions

In conclusion, in our study, tetanic stimulation did not shorten the time to stabilization of the muscle response, change the initial or final T1 height, or the onset time of action of rocuronium and did not change the pharmacodynamics of rocuronium in the clinical duration: the time to achieve a TOF value of 90% and 100% (recovery) or other TOF recovery parameters in patients aged 60 to 80 years.

## Supporting information

**S1 Dataset. Data presented in this paper.**
(DOCX)

**S1 Checklist. CONSORT 2010 checklist.**
(DOC)

**S1 Protocol. Clinical study protocol.**
(DOCX)

## Acknowledgments

We are grateful for the support of the Federal University Fluminense, Rio de Janeiro, Brazil; Department of Surgery, Pedro Ernesto University Hospital, affiliated with Estado do Rio de Janeiro University, Rio de Janeiro, Brazil; and Gaffrée & Guinle Federal University Hospital. The authors received no specific funding for this work.

## Author Contributions

**Conceptualization:** Angela Henrique Silva Ribeiro, Ismar Lima Cavalcanti.

**Data curation:** Angela Henrique Silva Ribeiro.

**Formal analysis:** Angela Henrique Silva Ribeiro.

**Investigation:** Nubia Verçosa, Estêvão Luiz Carvalho Braga.

**Methodology:** Angela Henrique Silva Ribeiro, Estêvão Luiz Carvalho Braga.

**Project administration:** Ismar Lima Cavalcanti.

**Validation:** Angela Henrique Silva Ribeiro, Angelo Jorge Queiroz Rangel Micuci, Paulo Alípio Germano-Filho.

**Writing – original draft:** Angela Henrique Silva Ribeiro, Nubia Verçosa, Ismar Lima Cavalcanti.

**Writing – review & editing:** Angela Henrique Silva Ribeiro, Nubia Verçosa, Angelo Jorge Queiroz Rangel Micuci, Paulo Alípio Germano-Filho, Ismar Lima Cavalcanti.

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
