## [Decision Letter · Decision Letter 0]

10 Jan 2024

PONE-D-23-40644Effect of tetanic stimulation prior to train-of-four monitoring on the time to muscle response stabilization in patients aged 60 to 80 years: A prospective randomized controlled trialPLOS ONE

Dear Dr. Henrique Silva Ribeiro,

Thank you for submitting your manuscript to PLOS ONE. After careful consideration, we feel that it has merit but does not fully meet PLOS ONE’s publication criteria as it currently stands. Therefore, we invite you to submit a revised version of the manuscript that addresses the points raised during the review process.

The study presented is a multicenter, prospective, randomized, single-blind, controlled trial with a 2-arm parallel group design, aiming to assess the time to stabilization of muscle response with tetanic stimulation in patients aged 60 to 80 years. The authors concluded that tetanic stimulation did not impact the time to T1 twitch stabilization in this age group. However, the manuscript has several deficiencies according to the CONSORT checklist. Specific issues include the lack of information on protocol changes, incomplete reporting in the abstract, insufficient p-value precision, the suggestion of using a Linear Mixed Effects Model for data structure, unclear definition of "stabilization," missing details on randomization, inadequate sample size justification, unclear statistical models in tables, and the need for dose-response modeling for certain data. Additionally, some statements and sections require clarification and rephrasing. Overall, the manuscript requires thorough revisions to address these issues and improve its quality.

We look forward to receiving your revised manuscript.

Kind regards,

Lalit Gupta

Academic Editor

PLOS ONE

Journal Requirements:

Additional Editor Comments:

The study presented is a multicenter, prospective, randomized, single-blind, controlled trial with a 2-arm parallel group design, aiming to assess the time to stabilization of muscle response with tetanic stimulation in patients aged 60 to 80 years. The authors concluded that tetanic stimulation did not impact the time to T1 twitch stabilization in this age group. However, the manuscript has several deficiencies according to the CONSORT checklist. Specific issues include the lack of information on protocol changes, incomplete reporting in the abstract, insufficient p-value precision, the suggestion of using a Linear Mixed Effects Model for data structure, unclear definition of "stabilization," missing details on randomization, inadequate sample size justification, unclear statistical models in tables, and the need for dose-response modeling for certain data. Additionally, some statements and sections require clarification and rephrasing. Overall, the manuscript requires thorough revisions to address these issues and improve its quality.

Reviewers' comments:

Reviewer's Responses to Questions

**Comments to the Author**

1. Is the manuscript technically sound, and do the data support the conclusions?

Reviewer #1: Yes

Reviewer #2: Partly

2. Has the statistical analysis been performed appropriately and rigorously? 

Reviewer #1: Yes

Reviewer #2: I Don't Know

3. Have the authors made all data underlying the findings in their manuscript fully available?

Reviewer #1: Yes

Reviewer #2: No

4. Is the manuscript presented in an intelligible fashion and written in standard English?

Reviewer #1: Yes

Reviewer #2: No

5. Review Comments to the Author

Reviewer #1: This study provides useful details for neuromuscular monitoring that has not been investigated yet.

Two comments: rocuronium should be mentioned in the title.

The second does not influence the results but as there is considerable difference between plasma or effect-site in TIVA, in page 10, line 151 it should be mentioned whether you used Marsh plasma-site (I suppose) or effect-site concentrations.

Reviewer #2: The authors reported the results of multicenter, prospective, randomized, single-blind, controlled trial with a 2 arm parallel group design to determine the time to stabilization of the muscle response with tetanic stimulation compared to no stimulation in patients aged 60 to 80 year. They concluded that the use of tetanic stimulation to shorten the time to T1 height stabilization in neuromuscular monitoring did not alter the stabilization time of the T1 twitch in patients between 60 and 80 years old.

In general, the current presentation has deficits and needs a sound revisions.

general: Please comment on each point of the CONSORT checklist in the manuscript. E.g. if no changes to the protocol are made, please state this explicitly.

L27-32: Please give the information required in the CONSORT abstract form here, e.g. setting, primary endpoint variable, randomization technique, etc.

L34ff: Please give p-values with at least 3 decimal digits.

L36-39: data seem to follow a "repeated" measures type of structure, which should be reflected in the analysis by a Linear Mixed Effects Model.

L76-80: First as biostatistician I am not sure about the objective definition of stabilization, which is necessary to determine the time. In particular, it is unclear, what is meant by " stabilization", whether " stabilization" was observed in every patient, whether the "degree of stabilization" is the same with every patient. Could you please provide more information.

Second, the specific hypothesis to be proven is not given.

L98-101: As enrollment starts at different times in the two centers, it is likely to assume a time trend in the data. So please provide at least a figure of the treatment effect by center to make clear that the treatment effect is consistent across centers.

L120-125: Please give details of the randomization. Missing details cover the randomization algorithm (e.g. permuted block randomization with block size 4?). It is unclear, how the random allocation is implemented, e.g. by the sealed opaque envelops (?), and who provided the random lots, who assigned patients to make concealment of randomization clear.

The technique to conceal and protect the information about the allocation from the patients (blinding) need to be described.

L199-203: Sample size justification need further information. The statistical test underlying the sample size justification is missing. Please state the software used for calculation. The result might be to low, as - assuming a two sample t-Test - a sample size of 26 / group is necessary without 20% dropout. Please note, that a dropout rate of 20% is rather large and may cause the validity. Please comment on these points in the limitation section, as the study might appear as underpowered.

L204ff: Please make the section specific to the content so to avoid general description and state which method is used in which situation. Give the statistical analysis of the primary endpoint variable. Give the analysis population and state how to deal with missing values to avoid attrition bias. Use the ITT population.

Pretest on normal distribution might be appropriate for explorative analysis, i.e. ancillary analysis.

L255: The basis should be 30 patients in each group according to flow chart. Statistical test for baseline patient characteristics are not meaningful in randomized clinical trials, delete last column of table 1.

L235: If MANOVA is applied there is more information needed to understand the statistical model. I assume the first row "Time .." just refers to the t-Test (underlying the sample size justification) and thus the p-value is from a t-Test. Please give means and SD for each group followed by a 95% confidence interval for mean difference. How are missing values dealed with. Last row could be deleted.

L244: Unclear what % means in row 2 and 3. With % numbers the column entries are difficult to understand. May bet this is an ANCOVA model.

Like in Table 2 MANOVA model not clear, please give the within and between factors, correlation structure and other model parameters.

L256: Data for "Time to a normalized TOF value of" seem to be a dose response profile and should be modelled accordingly to establish differences.

L338: Statement is not clear and need rephrasing.

6. PLOS authors have the option to publish the peer review history of their article (what does this mean?). If published, this will include your full peer review and any attached files.

Reviewer #1: No

Reviewer #2: No

---

## [Author Response · Author response to Decision Letter 0]

21 Feb 2024

Lalit Gupta

Academic Editor

PLOS ONE

RE: PONE-D-23-40644, entitled "Effect of tetanic stimulation prior to train-of-four monitoring on the time to muscle response stabilization in patients aged 60 to 80 years: A prospective randomized controlled trial"

Dear Dr. Lalit Gupta

Thank you very much for your letter dated January 10, 2024, and for the careful review of our manuscript, which we have amended following the reviewers’ suggestions. A marked-up copy of the revised manuscript (with track changes) as well as an unmarked copy of the manuscript (without track changes) have been uploaded to the submission system. Also, please find below an itemized point-by-point response to the reviewers’ comments.

We aimed in this study at the hypothesis that tetanic stimulation would shorten the stabilization time in neuromuscular blockade monitoring in patients aged 60 to 80 years and interfere on the onset time of rocuronium and recovery times, but that was not proven in our study. 

We look forward to hearing from you about the status of our manuscript, which we hope is now acceptable for publication in PLOS ONE. Please feel free to contact me if you require any additional information. 

Sincerely,

Angela Henrique S. Ribeiro

angelahenriqueanest@gmail.com

¬------------------------------------------------------------

Response: No changes were made to our financial disclosure.

If applicable, we recommend that you deposit your laboratory protocols in protocols.io to enhance the reproducibility of your results. Protocols.io assigns your protocol its own identifier (DOI) so that it can be cited independently in the future. 

Response: We thank you for introducing us to protocols.io. However, our data are deposited in Mendeley Database Repository and the original protocol was kept unchanged from the beginning. “Ribeiro AHS, Germano-Filho PA, Cavalcanti I, Figueiredo NV (2022). Effect of tetanic stimulation prior to train-of-four sequence on stabilization time of muscle response in patients aged 60-80. Mendeley Data, V1, doi:10.17632/fyhpfpk5cm.1”

Journal Requirements:

Additional Editor Comments:

The study presented is a multicenter, prospective, randomized, single-blind, controlled trial with a 2-arm parallel group design, aiming to assess the time to stabilization of muscle response with tetanic stimulation in patients aged 60 to 80 years. The authors concluded that tetanic stimulation did not impact the time to T1 twitch stabilization in this age group. However, the manuscript has several deficiencies according to the CONSORT checklist. Specific issues include the lack of information on protocol changes, incomplete reporting in the abstract, insufficient p-value precision, the suggestion of using a Linear Mixed Effects Model for data structure, unclear definition of "stabilization," missing details on randomization, inadequate sample size justification, unclear statistical models in tables, and the need for dose-response modeling for certain data. Additionally, some statements and sections require clarification and rephrasing. Overall, the manuscript requires thorough revisions to address these issues and improve its quality.

Response: Thank you for the comments, we altered the article and made the corrections according to your guidance.

Comments to the Author

1. Is the manuscript technically sound, and do the data support the conclusions?

Reviewer #1: Yes

Reviewer #2: Partly

2. Has the statistical analysis been performed appropriately and rigorously?

Reviewer #1: Yes

Reviewer #2: I Don't Know

3. Have the authors made all data underlying the findings in their manuscript fully available?

Reviewer #1: Yes

Reviewer #2: No

Response: We thank you for your questions and suggestions. All data underlying the findings described in the manuscript are fully available without restriction. The data collected for the research were deposited in a public repository, Mendeley Database Repository: “Ribeiro AHS, Germano-Filho PA, Cavalcanti I, Figueiredo NV (2022). Effect of tetanic stimulation prior to train-of-four sequence on stabilization time of muscle response in patients aged 60-80”. Mendeley Data, V1, doi:10.17632/fyhpfpk5cm.1.”

The data points behind means, medians and variance measures are deposited in this public repository. There are no restrictions on publicly sharing data.

4. Is the manuscript presented in an intelligible fashion and written in standard English?

Reviewer #1: Yes

Reviewer #2: No

Response: We are not native English speakers, so we used the services of a specialized company, American Journal Experts (AJE) – Springer Nature, registered SKLF32XNL, Sep 2023. We have uploaded AJE’s Editorial Certificate to the submission system as Supplementary Information.

5. Review Comments to the Author

Reviewer #1: This study provides useful details for neuromuscular monitoring that has not been investigated yet.

Two comments: rocuronium should be mentioned in the title.

Response: Thank you for your considerations and suggestions. We included rocuronium in the title as suggested (page 1, lines 1-3).

The second does not influence the results but as there is considerable difference between plasma or effect-site in TIVA, in page 10, line 151 it should be mentioned whether you used Marsh plasma-site (I suppose) or effect-site concentrations.

Response: We used Marsh pharmacokinetic model in this research. Included in the manuscript (page 10, line 161).

Reviewer #2: The authors reported the results of multicenter, prospective, randomized, single-blind, controlled trial with a 2 arm parallel group design to determine the time to stabilization of the muscle response with tetanic stimulation compared to no stimulation in patients aged 60 to 80 year. They concluded that the use of tetanic stimulation to shorten the time to T1 height stabilization in neuromuscular monitoring did not alter the stabilization time of the T1 twitch in patients between 60 and 80 years old.

In general, the current presentation has deficits and needs a sound revisions.

general: Please comment on each point of the CONSORT checklist in the manuscript. E.g. if no changes to the protocol are made, please state this explicitly.

Response: Thank you for your considerations and suggestions. Each point of the CONSORT checklist was revised as requested, and the suggested revisions were made to the manuscript. No changes to the anesthesia protocol or methods were made (page 7, lines 92-93). We just added another site center due to difficulty recruiting during the pandemic.

L27-32: Please give the information required in the CONSORT abstract form here, e.g. setting, primary endpoint variable, randomization technique, etc.

Response: The information required in the CONSORT abstract form was provided in the revised manuscript (page 3, lines 26-36). 

L34ff: Please give p-values with at least 3 decimal digits.

Response: P-values with at least 3 decimal digits were provided throughout the revised manuscript. 

L36-39: data seem to follow a "repeated" measures type of structure, which should be reflected in the analysis by a Linear Mixed Effects Model.

Response: The measurements were not repeated but rather consecutive. The neuromuscular blockade recovery times measured are not comparable. TOF 40 is muscle strength recovery of 40% of baseline, TOF 50 is muscle strength recovery of 50% of baseline, TOF 60 is muscle strength recovery of 60% of baseline, TOF 70 is muscle strength recovery of 70% of baseline, TOF 80 is muscle strength recovery of 80% of baseline, TOF 90 is muscle strength recovery of 90% of baseline, up to total muscle strength recovery of 100% of TOF.

L76-80: First as biostatistician I am not sure about the objective definition of stabilization, which is necessary to determine the time. In particular, it is unclear, what is meant by " stabilization", whether " stabilization" was observed in every patient, whether the "degree of stabilization" is the same with every patient. Could you please provide more information.

Response: According to the Stockholm Revision (ref 7), muscle response stabilization is defined as a variation of less than 5% in T1 height (first twitch) in response to the electrical stimulus applied to the corresponding nerve. The stabilization time used in this study was defined as the period of time between the end of calibration (adjusted twitch height to 100%) and achievement of a muscle response variation (T1-first twitch) of less than 5% for at least 2 minutes (page 6, lines 69-70). The same stabilization concept was used for all patients. The time to reach stabilization is individual, so each patient will have their own time to stabilize the muscle response to the electrical stimulus. The characteristics of the electrical stimulus that all patients received were standardized; what varied was the time for the muscle response to stabilize, as this is an individual organic response for each patient. The neuromuscular blockade is monitored by applying a small electric shock in a sequence of four stimuli, leading to four muscle contractions, known as train-of-four (TOF). The electric shock to the peripheral nerve sends a signal for the corresponding muscle to contract. In this study, we used stimulation of the ulnar nerve to contract the adductor pollicis (page 10, lines 149-151; page 11, lines 164-165).

¬¬¬¬¬¬¬¬¬¬¬

Second, the specific hypothesis to be proven is not given.

Response: The hypothesis was added to the manuscript (page 6, lines 77-78). Our hypothesis was that the use of tetanic stimulus prior to the sequence of four stimuli would decrease the stabilization time of muscular response and alter the onset and recovery times of the neuromuscular blockade.

L98-101: As enrollment starts at different times in the two centers, it is likely to assume a time trend in the data. So please provide at least a figure of the treatment effect by center to make clear that the treatment effect is consistent across centers.

Response: Enrollment started at different times in the two centers, but there was no time trend in the data. The measurements were independent and individual, starting and ending during the same anesthetic-surgical procedure. There was no time effect on the data.

L120-125: Please give details of the randomization. Missing details cover the randomization algorithm (e.g. permuted block randomization with block size 4?). It is unclear, how the random allocation is implemented, e.g. by the sealed opaque envelops (?), and who provided the random lots, who assigned patients to make concealment of randomization clear. The technique to conceal and protect the information about the allocation from the patients (blinding) need to be described.

Response: The randomization was made with 60 random numbers generated in the computer program WinPepi (Program for Epidemiologists for Windows) for the two groups. The study participants were not aware of the allocated group, as they were anesthetized, unconscious, and therefore blinded to the allocation. We obeyed the random allocation sequence generated by the program. The random allocation was implemented by sealed opaque envelopes, which were opened by a single research anesthesiologist after anesthesia induction. Blinding at this point was not possible. The technique to conceal and protect the information about the allocation from the patients was detailed in the revised manuscript, as suggested by the reviewer (page 9, lines 127-135).

L199-203: Sample size justification need further information. The statistical test underlying the sample size justification is missing. Please state the software used for calculation. The result might be to low, as - assuming a two sample t-Test - a sample size of 26 / group is necessary without 20% dropout. Please note, that a dropout rate of 20% is rather large and may cause the validity. Please comment on these points in the limitation section, as the study might appear as underpowered.

Response: Student’s t-test was used to calculate the sample size (page 13, line 211). To use this test, it was necessary to determine the expected effect size (variable: electrical signal stabilization time), which is the ratio of the difference between the means of the two groups to the common standard deviation of the difference. By setting an effect size of 0.8 standard deviations, we estimated this effect for the underlying test. The means for the intervention and control groups were 2.5 (SD, 1.0) and 5.0 (SD, 5.2), respectively, in the pilot study with 10 patients, 5 in each group. Selection bias should be avoided, but a 20% loss is usually acceptable in studies, as reported in the following article: “Schulz KF, Grimes DA. Sample size slippages in randomized trials: exclusions and the lost and wayward. Lancet 2002; 359(9308): 781-5.”

We included the dropout rate in the limitations section (page 22, line 344-345).

L204ff: Please make the section specific to the content so to avoid general description and state which method is used in which situation. Give the statistical analysis of the primary endpoint variable. Give the analysis population and state how to deal with missing values to avoid attrition bias. Use the ITT population.

Pretest on normal distribution might be appropriate for explorative analysis, i.e. ancillary analysis.

Response: We stated which method is used in each situation and gave the statistical analysis of the primary endpoint variable in the revised manuscript. The data were normally distributed according to the Shapiro-Wilk test. Student’s t test was used to compare normally distributed continuous variables (age, BMI, and creatinine clearance) between the intervention and control groups. For categorical variables, Pearson’s chi-square test or Fisher's exact test was used to compare proportions (sex and ASA class) between the intervention and control groups. Due to the

---

## [Decision Letter · Decision Letter 1]

8 Mar 2024

PONE-D-23-40644R1Effect of tetanic stimulation prior to train-of-four monitoring on the time to muscle response stabilization with neuromuscular blockade with rocuronium in patients aged 60 to 80 years: A prospective randomized controlled trialPLOS ONE

Dear Dr. Henrique Silva Ribeiro,

Thank you for submitting your manuscript to PLOS ONE. After careful consideration, we feel that it has merit but does not fully meet PLOS ONE’s publication criteria as it currently stands. Therefore, we invite you to submit a revised version of the manuscript that addresses the points raised during the review process.

The study aimed to determine the time to stabilization of muscle response with tetanic stimulation compared to no stimulation in patients aged 60 to 80 years. However, the use of tetanic stimulation did not alter the stabilization time of the T1 twitch in this age group. Several points need clarification, including the definition of “stabilization,” specific hypotheses, and randomization details. Additionally, addressing the time trend across centers would enhance the analysis.

We look forward to receiving your revised manuscript.

Kind regards,

Lalit Gupta

Academic Editor

PLOS ONE

Additional Editor Comments:

The study aimed to determine the time to stabilization of muscle response with tetanic stimulation compared to no stimulation in patients aged 60 to 80 years. However, the use of tetanic stimulation did not alter the stabilization time of the T1 twitch in this age group. Several points need clarification, including the definition of “stabilization,” specific hypotheses, and randomization details. Additionally, addressing the time trend across centers would enhance the analysis.

Reviewers' comments:

Reviewer's Responses to Questions

**Comments to the Author**

1. If the authors have adequately addressed your comments raised in a previous round of review and you feel that this manuscript is now acceptable for publication, you may indicate that here to bypass the “Comments to the Author” section, enter your conflict of interest statement in the “Confidential to Editor” section, and submit your "Accept" recommendation.

Reviewer #1: (No Response)

Reviewer #2: (No Response)

2. Is the manuscript technically sound, and do the data support the conclusions?

Reviewer #1: (No Response)

Reviewer #2: Partly

3. Has the statistical analysis been performed appropriately and rigorously? 

Reviewer #1: (No Response)

Reviewer #2: No

4. Have the authors made all data underlying the findings in their manuscript fully available?

Reviewer #1: (No Response)

Reviewer #2: No

5. Is the manuscript presented in an intelligible fashion and written in standard English?

Reviewer #1: (No Response)

Reviewer #2: Yes

6. Review Comments to the Author

Reviewer #1: (No Response)

Reviewer #2: L77: The hypotheses refers to three objectives = criteria or primary endpoint variables, which is not in line with the methodological approach. Thus only one variable should be mentioned " We hypothesized that the tetanic stimulation could decrease stabilization time." .

L211: Sample size justification is not correct. The provided sample size calculation relies on the z-Test rather than on the t-Test. Conditioned on the assumed effect size, the dropout rate needs to be added, which results in a total sample size of 62.5 in case of the z-test and 65 in case of the t-test. Thus, in the limitation section this needs to be stated resulting in the conclusion, that the study is underpowered.

L204ff: In the statistical method section delete the test on normal distribution, as the test is not meaningfull for analysis of the primary endpoint.

The analysis population is the Per Protocol population.

Please state in the limitation section, that the analysis population, as per protocol, results in anticonservatve test decision.

7. PLOS authors have the option to publish the peer review history of their article (what does this mean?). If published, this will include your full peer review and any attached files.

Reviewer #1: No

Reviewer #2: No

---

## [Author Response · Author response to Decision Letter 1]

8 Apr 2024

Lalit Gupta

Academic Editor

PLOS ONE

RE: PONE-D-23-40644R1, entitled "Effect of tetanic stimulation prior to train-of-four monitoring on the time to muscle response stabilization with neuromuscular blockade with rocuronium in patients aged 60 to 80 years: A prospective randomized controlled trial"

Dear Dr. Lalit Gupta

Thank you for your email dated March 8, 2024, and once again for the careful review of our manuscript, which we have amended following the reviewer’s suggestions. A marked-up copy of the revised manuscript (with track changes) as well as an unmarked copy of the manuscript (without track changes) have been uploaded to the submission system. Also, please find below an itemized point-by-point response to the reviewers’ comments.

We look forward to hearing from you about the status of our manuscript, which we hope is now acceptable for publication in PLOS ONE. Please feel free to contact me if you require any additional information. 

Sincerely,

Angela Henrique S. Ribeiro

angelahenriqueanest@gmail.com

Additional Editor Comments:

The study aimed to determine the time to stabilization of muscle response with tetanic stimulation compared to no stimulation in patients aged 60 to 80 years. However, the use of tetanic stimulation did not alter the stabilization time of the T1 twitch in this age group. Several points need clarification, including the definition of “stabilization,” specific hypotheses, and randomization details. Additionally, addressing the time trend across centers would enhance the analysis.

Response: Thank you for your considerations. We clarified those points in the text as required. More specifically, the definition of “stabilization” is fully explained in the Stockholm Revision Guidelines for researchers (ref 7 of the manuscript), and our study strictly followed the guidelines. We clarified in the revised manuscript the definition of stabilization, as defined by the guidelines (page 6, lines 69-70). Also, the specific hypotheses were altered as requested by the reviewer (page 6, lines 77-78). Randomization details were added in the previous round as requested by the reviewers, and complemented in this round (page 9, lines 127-136). Regarding the issue about time trend across centers, raised in the previous round of review, we explained that, although enrollment started at different times in the two centers, there was no time trend in the data because the measurements were made independently and individually, starting and ending in the same anesthetic-surgical procedure. Therefore, there was no time effect on the data. 

Reviewers' comments:

Reviewer's Responses to Questions

Comments to the Author

1. If the authors have adequately addressed your comments raised in a previous round of review and you feel that this manuscript is now acceptable for publication, you may indicate that here to bypass the “Comments to the Author” section, enter your conflict of interest statement in the “Confidential to Editor” section, and submit your "Accept" recommendation.

Reviewer #1: (No Response)

Reviewer #2: (No Response)

2. Is the manuscript technically sound, and do the data support the conclusions?

Reviewer #1: (No Response)

Reviewer #2: Partly

3. Has the statistical analysis been performed appropriately and rigorously?

Reviewer #1: (No Response)

Reviewer #2: No

4. Have the authors made all data underlying the findings in their manuscript fully available?

Reviewer #1: (No Response)

Reviewer #2: No

5. Is the manuscript presented in an intelligible fashion and written in standard English?

Reviewer #1: (No Response)

Reviewer #2: Yes

6. Review Comments to the Author

Reviewer #1: (No Response)

Reviewer #2: L77: The hypotheses refers to three objectives = criteria or primary endpoint variables, which is not in line with the methodological approach. Thus only one variable should be mentioned " We hypothesized that the tetanic stimulation could decrease stabilization time." .

Response: Thank you for your considerations. We changed the text as suggested, with just one variable mentioned (page 6, lines 77-78).

L211: Sample size justification is not correct. The provided sample size calculation relies on the z-Test rather than on the t-Test. Conditioned on the assumed effect size, the dropout rate needs to be added, which results in a total sample size of 62.5 in case of the z-test and 65 in case of the t-test. Thus, in the limitation section this needs to be stated resulting in the conclusion, that the study is underpowered.

Response: Thank you for your considerations. Due to the paucity of data in the literature, the sample size was calculated based on a pilot study with 10 patients. Considering a significance level of 5% and a power of 80% to detect a reduction in the time to stabilization of the expected T1 height within 8 minutes after the tetanic stimulation, an estimated standard deviation of 10 minutes (minimum effect size of 0.8 standard deviations) and an estimated 20% of potential losses, a sample size of 30 patients per group was needed (page 13, lines 211-215). If we had achieved an effect size of 0.8 standard deviations, we would have found statistical significance, but the effect size found in the study was 0.5. Therefore, the “n” was not sufficient because the effect found was smaller than that initially obtained in the pilot study and not because of an error in the sample size calculation. As requested, we included this explanation in the limitation section (page 22, lines 344-348).

L204ff: In the statistical method section delete the test on normal distribution, as the test is not meaningfull for analysis of the primary endpoint.

Response: Thank you for your considerations. We excluded from the statistical method section the test on normal distribution as required (page 14, line 220).

The analysis population is the Per Protocol population.

Please state in the limitation section, that the analysis population, as per protocol, results in anticonservatve test decision.

Response: Thank you for your consideration. As required, we included in the limitation section that the analysis population, as per protocol, resulted in anticonservative test decision (page 22, line 346).

7. PLOS authors have the option to publish the peer review history of their article (what does this mean?). If published, this will include your full peer review and any attached files.

Do you want your identity to be public for this peer review? For information about this choice, including consent withdrawal, please see our Privacy Policy.

Reviewer #1: No

Reviewer #2: No

---

## [Decision Letter · Decision Letter 2]

22 Apr 2024

Effect of tetanic stimulation prior to train-of-four monitoring on the time to muscle response stabilization with neuromuscular blockade with rocuronium in patients aged 60 to 80 years: A prospective randomized controlled trial

PONE-D-23-40644R2

Dear Dr. Henrique Silva Ribeiro,

We’re pleased to inform you that your manuscript has been judged scientifically suitable for publication and will be formally accepted for publication once it meets all outstanding technical requirements.

Kind regards,

Lalit Gupta

Academic Editor

PLOS ONE

Additional Editor Comments (optional):

Al the major changes asked have been done by Authors and are acknowledged in further review.

Reviewers' comments:

Reviewer's Responses to Questions

**Comments to the Author**

1. If the authors have adequately addressed your comments raised in a previous round of review and you feel that this manuscript is now acceptable for publication, you may indicate that here to bypass the “Comments to the Author” section, enter your conflict of interest statement in the “Confidential to Editor” section, and submit your "Accept" recommendation.

Reviewer #2: All comments have been addressed

2. Is the manuscript technically sound, and do the data support the conclusions?

Reviewer #2: Yes

3. Has the statistical analysis been performed appropriately and rigorously? 

Reviewer #2: Yes

4. Have the authors made all data underlying the findings in their manuscript fully available?

Reviewer #2: Yes

5. Is the manuscript presented in an intelligible fashion and written in standard English?

Reviewer #2: Yes

6. Review Comments to the Author

Reviewer #2: All comments are adressed, The paper is ready for publication.

7. PLOS authors have the option to publish the peer review history of their article (what does this mean?). If published, this will include your full peer review and any attached files.

Reviewer #2: No

---

## [Editor Report · Acceptance letter]

24 May 2024

PONE-D-23-40644R2 

PLOS ONE

Dear Dr. Henrique Silva Ribeiro, 

I'm pleased to inform you that your manuscript has been deemed suitable for publication in PLOS ONE. Congratulations! Your manuscript is now being handed over to our production team.

Kind regards, 

on behalf of

Dr. Lalit Gupta 

Academic Editor

PLOS ONE